# Understanding COVID-19 Vaccine Effectiveness against Death Using a Novel Measure: COVID Excess Mortality Percentage

**DOI:** 10.3390/vaccines11020379

**Published:** 2023-02-07

**Authors:** Vladimir Atanasov, Natalia Barreto, Jeff Whittle, John Meurer, Benjamin W. Weston, Qian (Eric) Luo, Lorenzo Franchi, Andy Ye Yuan, Ruohao Zhang, Bernard Black

**Affiliations:** 1Mason College of Business, William & Mary, Williamsburg, VA 23185, USA; 2Department of Economics, University of Illinois Urbana-Champaign, Champaign, IL 61820, USA; 3Medical College of Wisconsin, Milwaukee, WI 53226, USA; 4Department of Health Policy and Management, George Washington University, Washington, DC 20052, USA; 5Pritzker School of Law, Northwestern University, Chicago, IL 60611, USA; 6Department of Data Science, Centre College, Danville, KY 40422, USA

**Keywords:** COVID-19, COVID-19 mortality, cause of death, COVID Excess Mortality Percentage, vaccine effectiveness, selection bias

## Abstract

COVID-19 vaccines have saved millions of lives; however, understanding the long-term effectiveness of these vaccines is imperative to developing recommendations for booster doses and other precautions. Comparisons of mortality rates between more and less vaccinated groups may be misleading due to selection bias, as these groups may differ in underlying health status. We studied all adult deaths during the period of 1 April 2021–30 June 2022 in Milwaukee County, Wisconsin, linked to vaccination records, and we used mortality from other natural causes to proxy for underlying health. We report relative COVID-19 mortality risk (RMR) for those vaccinated with two and three doses versus the unvaccinated, using a novel outcome measure that controls for selection effects. This measure, COVID Excess Mortality Percentage (CEMP), uses the non-COVID natural mortality rate (Non-COVID-NMR) as a measure of population risk of COVID mortality without vaccination. We validate this measure during the pre-vaccine period (Pearson correlation coefficient = 0.97) and demonstrate that selection effects are large, with non-COVID-NMRs for two-dose vaccinees often less than half those for the unvaccinated, and non-COVID NMRs often still lower for three-dose (booster) recipients. Progressive waning of two-dose effectiveness is observed, with an RMR of 10.6% for two-dose vaccinees aged 60+ versus the unvaccinated during April–June 2021, rising steadily to 36.2% during the Omicron period (January–June, 2022). A booster dose reduced RMR to 9.5% and 10.8% for ages 60+ during the two periods when boosters were available (October–December, 2021; January–June, 2022). Boosters thus provide important additional protection against mortality.

## 1. Introduction

COVID-19 vaccines have saved millions of lives, and mortality rates have fallen substantially in the second half of 2022 versus 2020, 2021, or the first half of 2022. However, U.S. mortality is running at an annual rate of over 100,000, and worldwide mortality is a large multiple of this rate. It is important to understand the mortality risk faced by the vaccinated, how protection varies with age, time since vaccination and other factors, and the benefit of booster doses.

Many studies have reported evidence on vaccine effectiveness (VE) against infection, hospitalization, and death [1,2]. However, most of these studies face potential confounding due to selection bias. Most studies on vaccine effectiveness against mortality (VE) have had limited controls for individual characteristics, often only age and gender [3,4,5,6,7,8,9], or study short time periods [5,10,11,12]. U.S.-based studies typically lacked population-level data [7,13,14,15], and studies with population-level data often have had limited information about health status [3,4,8,9,16]. Some studies used a test-negative design [10,14,15], which is prone to selection bias [17]. If healthier people are more likely to be vaccinated, lower mortality rates for vaccinees may partly reflect their (unobserved) better health and thus lower inherent risk of mortality.

In this study, we propose a novel method for evaluating COVID-19 vaccine effectiveness against mortality at the population level, which uses non-COVID natural mortality rates as a surrogate for underlying health. This surrogate is attractive because non-COVID mortality, COVID mortality, and vaccination status are often available for the same population. We propose, as an outcome measure, the COVID Excess Mortality Percentage (CEMP), defined as COVID-19 deaths divided by non-COVID natural deaths, converted to a percentage. The CEMP denominator controls for differences in population health between two groups, such as vaccinated versus unvaccinated.

We provide evidence on selection bias, measure the magnitude of this bias, and validate the CEMP measure as a means of controlling for selection bias. We then use CEMP as an outcome to measure VE against death for the entire adult population of a large U.S. city. We compare relative mortality risk (RMR = 1 − VE) for two- and three-dose vaccinees versus the unvaccinated, computed using this measure, to previous approaches that often lack robust adjustment for underlying health status.

## 2. Data and Methods

We obtained linked, de-identified mortality and vaccination records for all adults aged 18+ in Milwaukee County, Wisconsin (adult population 722,000) for 1 January 2021 through 30 June 2022, including residence zip code, age at death, gender, race/ethnicity, education, income, marital status, veteran status, manner of death, and text fields for cause of death, and conditions leading to or contributing to death. See Appendix A for sample details.

We used text analysis to identify deaths due to COVID-19 versus other natural causes. This approach counts more COVID-19 deaths than relying on ICD-10 cause-of-death codes prepared by the National Center for Health Statistics (NCHS), based on the text fields. Our approach was developed before we had access to ICD-10 codes; we sought to develop a reasonable method for estimating COVID-19 deaths based on the text fields. Our approach identified 1934 COVID deaths versus 1369 using ICD-10 codes (71% of our count). A major difference was that NCHS coded many COVID deaths as B99 (other and unspecified infectious diseases) instead of as U07.1 (COVID) (Appendix A). In our judgment, most of the cases that we identify as COVID-19, but the NCHS does not, or vice versa, are not close cases. For Wisconsin as a whole, the percentage undercount using ICD-10 codes from NCHS is smaller; we counted 12,595 COVID-19 deaths versus 11,512 with NCHS coding (92% of our count). Further details on our text-based algorithm can be found in the Appendix A.

We treat vaccination as effective against mortality starting 30 days after receipt; this allows for a vaccine dose to become fully effective, as well as the typical several-week lag between infection and death. We exclude immune-compromised decedents.

The mRNA vaccines (from Moderna, mRNA1273; and Pfizer-BioNTech BNT162b2) use two initial does; J&J uses one dose. We report results based on number of doses, thus treating one J&J dose as equivalent to one mRNA dose, but we obtain similar results if we exclude J&J vaccinees. Standard U.S. two-dose timing was four weeks between doses for Moderna and three weeks for Pfizer. Mixing of vaccine types was uncommon (Appendix A).

We define CEMP, VE versus the unvaccinated, and relative mortality risk after vaccination (RMR) in each time period, within a population group, as:$$\mathrm{CEMP}=\frac{\mathrm{COVID}\;\mathrm{deaths}}{\mathrm{non}-\mathrm{COVID}\;\mathrm{natural}\;\mathrm{deaths}}$$
$$\mathrm{VE}=\frac{\left({\mathrm{CEMP}}_{\mathrm{unvax}}-{\mathrm{CEMP}}_{\mathrm{vax}}\right)\;}{{\mathrm{CEMP}}_{\mathrm{unvax}}}$$
$$\mathrm{RMR}=1-\mathrm{VE}=\frac{{\mathrm{CEMP}}_{\mathrm{vax}}\;}{{\mathrm{CEMP}}_{\mathrm{unvax}}}$$

RMR can be obtained by comparing mortality rates for two groups, or as an odds ratio from logistic regression for a population containing both groups. We also compute RMR and VE for two-versus-one-dose and three-versus-two-dose vaccinees. CEMP, VE, and RMR involve ratios, so they could be undefined if the denominator is zero; we did not have this issue with our data and population groups.

CEMP represents the odds, for natural-cause decedents, of dying from COVID-19 versus other natural causes. The ratio of CEMPs for two groups, such as vaccinated versus unvaccinated, is an odds ratio, obtainable from logistic regression. We conducted simple comparisons of CEMP for two groups, defined by age and vaccination status (and, in the Appendix A, also gender and race/ethnicity), and also conducted multivariate logistic regression analysis of the association between vaccination and RMR, in which we adjust for other variables, available from death certificates, that may be associated with mortality risk. The predictors in the regression analysis were age, age-squared, zip-code-level socio-economic status (zip-SES), gender, race/ethnicity, education level, marital status, and military veteran status.

We measured race/ethnicity as non-Hispanic White (“White”), Black, non-Black Hispanic (“Hispanic”), and Other (including Asian, Native American, and mixed race). We measured zip-SES using the Graham Social Deprivation Index, which we found to perform well in another work [18]. We estimated population in 2020 from the American Community Survey, used our vaccination data to measure the number of vaccinees; and assumed other persons were unvaccinated. We measure the non-COVID-19 natural mortality rate (non-COVID-NMR or NCNMR) for a group as non-COVID natural deaths divided by population.

CEMP treats the non-COVID natural mortality rate as a proxy for overall health of a given group, and thus the likelihood of mortality if not vaccinated. We assessed the validity of this approach out-of-sample, by studying the correlation in Indiana (a nearby state where we have mortality data), between natural mortality in April–December 2019 (pre-COVID) and COVID-19 mortality over April–December 2020 (the same months during the pre-vaccine period) Using 2019 natural mortality (rather than 2020 non-COVID natural mortality) to predict 2020 COVID mortality avoids the mechanical correlation which could arise if COVID deaths are undercounted or prior COVID infection leads to higher non-COVID natural mortality.

Our RMR estimates would be biased only if both: (i) COVID-19 mortality was undercounted, and (ii) the degree of undercounting differed systematically between vaccinated and unvaccinated persons. We assessed condition (i)—whether significant undercounting exists—as follows. We used natural mortality in Wisconsin over 2017–2019 to predict non-COVID natural mortality rates in the same month in 2020, using linear extrapolation, and compared predicted to measured non-COVID natural mortality during our sample period. We assessed whether measured non-COVID natural mortality exceeded predicted mortality, either overall, or during periods of high COVID mortality. Even if undercounting exists, we have no reason to expect that condition (ii) holds, but cannot provide evidence on this with our data.

This project was approved by the Medical College of Wisconsin Human Research Review Board.

## 3. Results

### 3.1. Validating the CEMP Measure

Figure 1 shows the correlation in Indiana between natural mortality in April–December 2019 (pre-COVID period) and COVID-19 mortality in April–December 2020 (the COVID period, but pre-vaccine) for population groups defined by age (the groups were 18–39, 40–49, 50–59, 60–69, 70–79, 80–89, and 90+), gender, race/ethnicity, and zip-SES. The Pearson correlation coefficient was 0.97, which provides evidence that non-COVID natural mortality strongly predicts COVID mortality for unvaccinated persons.

Further validation came from the multivariate logistic regression analysis discussed below. The RMR estimates within groups defined solely by age were similar to the multivariate estimates, which adjust for other factors that are associated with COVID-19 mortality. The similarity between the simpler estimates and multivariate estimates is consistent with the simpler CEMP measure already controlling effectively for population health.

### 3.2. Validating the Text-Based Measure of COVID-19 Deaths

U.S. national evidence suggests that there has been some undercounting of deaths attributable directly or indirectly to COVID-19 [19,20]. Bias in RMRs could result if COVID-19 deaths are undercounted and the degree of undercounting is associated with vaccination status. However, as shown in Figure 2, once we coded COVID-19 deaths based on analyzing the text fields in death certificates, we found no evidence of important undercounting, either in Milwaukee or in Wisconsin generally.

Figure 2 reports monthly natural non-COVID-19 and all natural deaths for Wisconsin for 2017–June 2022. For the pandemic period, we also show predicted natural non-COVID deaths, based on linear extrapolation from 2017–2019 to the same calendar month during the pandemic period. Natural deaths (including COVID-19 deaths) show two COVID-related peaks in late 2020 and late 2021–early 2022. Natural non-COVID-19 deaths had much smaller spikes, with magnitudes consistent with the usual tendency for mortality to rise in the winter. Predicted natural non-COVID natural deaths (dashed line) were close to measured non-COVID natural deaths, sometimes higher or lower, but with no obvious pattern. Measured non-COVID natural deaths were never above the 95% confidence interval (CI) for the predicted deaths (for CIs, see Appendix A).

This comparison, and the further analyses in Appendix A, provides evidence that our text-based coding of COVID-19 deaths performed well in capturing actual COVID-19 deaths, and thus provided reasonable counts for both the CEMP numerator and denominator.

Comparing actual to predicted non-COVID natural deaths can also address the possibility that vaccination is causing excess non-COVID deaths. If this occurred in substantial numbers, we would expect non-COVID natural deaths to be above predicted deaths, especially in early to mid-2021, when most people received their primary vaccination. This is not observed, either for all persons (Figure 2) or for persons aged 80+, who will be more frail and thus potentially more susceptible to any adverse effects of vaccination (Appendix A).

### 3.3. Evidence for Selection Effects

Table 1 provides evidence on differences in baseline health, using Non-COVID-NMR as a surrogate for health, between three-dose vaccinees, two-dose vaccinees, and the unvaccinated. The table reports Non-COVID-NMRs by age group, vaccination status, and time period, and the ratios of Non-COVID-NMR for two-dose and three-dose vaccinees to that for the unvaccinated. The table uses *’s to report statistical significance for the ratios relative to the null (100%), see Appendix A for 95% confidence intervals (CIs). Vaccinees had substantially lower non-COVID-NMRs, consistent with vaccinees being healthier on average, and thus likely facing lower baseline COVID mortality risk.

In the first two periods, before boosters were available, Non-COVID-NMRs for two-dose recipients were well below those for the unvaccinated, although differences were smaller for persons aged 80+. In the booster-available periods, there was a further separation, in which some persons who previously received two doses chose to obtain a third booster dose, while others did not. This choice also involves strong selection effects. Three-dose recipients had lower non-COVID-NMRs than two-dose recipients, with a smaller difference for ages 80+.

There is evidence from Table 1 that the non-COVID-NMR ratios of the vaccinated to the unvaccinated varied over time, and were generally higher in the Omicron period, especially for ages 60–79 and 80+. We discuss in the Appendix A possible explanations for this time variation.

Appendix A provides information on Milwaukee County vaccination rates and how they varied over time. These rates were broadly in line with national averages. Overall, around 74% of the adult population received at least one dose, 70% were fully vaccinated (one J&J dose or two mRNA doses), most of whom were two-dose recipients, and 56% received a third dose, with higher two-dose and three-dose rates for older persons.

### 3.4. CEMP and RMR by Time Period and Age Range: Overview

Table 2 reports the number of COVID-19 deaths, non-COVID-19 natural deaths, and CEMP (the ratio of the two), by age range and number of doses, for four periods: April–June 2021 (2Q-2021; Alpha as dominant variant); July–September 2021 (3Q-2021, Delta dominant, no boosters); October–December 2021 (4Q-2021, Delta dominant, boosters available); and January–June 2022 (1H-2022, Omicron dominant, boosters available). These periods correspond to when the respective variants accounted for most infections (see Appendix A).

Table 2 presents unadjusted results for COVID-19 deaths, non-COVID natural deaths, and CEMP by age group, time period, and vaccination status, and RMR for vaccinated versus unvaccinated persons. We present results by period, given evidence from other studies on waning vaccine effectiveness over time, differences in variant severity, and potential RMR differences between variants. For the combined age groups (18–59, 60+), the table uses *’s to report statistical significance for the ratios, see Appendix A for CIs.

CEMP levels for all groups were low in 2Q-2021—a relatively low period for COVID-19 infections and deaths—but rose substantially after that. For all unvaccinated persons, CEMP by period was 5.0%, 17.9%, 35.0%, and 18.6%. CEMP levels fell much more sharply during the Omicron period for persons aged 18–59 than for ages 60+.

### 3.5. RMR for Two Doses Versus the Unvaccinated

For the full sample, two-dose RMR levels by period rose from 10.6% to 17.3%, 21.1%, and 36.2%, implying decreased vaccine protection.

We found important differences in two-dose RMR for younger versus older persons. There was progressive waning for ages 60+, with RMR rising from 11.1% to 20.8%, 27.7%, and 34.2% by time period. For persons aged 18–59, the RMR by period was 0% (no deaths), 6.2%, and 3.3%, before rising to 42.7% in the Omicron period. Two-dose RMR was nearly zero for ages 18–39, with only one death—a severely comorbid 35-year-old woman during 1Q-2022.

### 3.6. RMR for Booster Dose

A booster dose offered considerable additional protection, especially for those aged 60+. RMR for booster recipients aged 60+ was 9.5% in 4Q-2021 and 10.8% in 1H-2022. Booster protection varied with age; for ages 18–59, booster RMR was 0% (no booster-recipient deaths versus. 100 unvaccinated deaths). Older persons, in contrast, had meaningful RMRs after a booster dose, although much lower than their two-dose RMRs.

Figure 3 shows RMR for two-dose elderly vaccinees by time period, and also RMR for three-dose vaccinees for the periods in which boosters were available. It includes separate lines for ages 60–79 and 80+. The figure shows rising RMRs over time (waning vaccine effectiveness) for both groups, as well as the higher RMRs for ages 80+.

### 3.7. RMR for One Dose

One-dose RMR versus the unvaccinated has been rarely studied. RMR relative to unvaccinated is substantial, at 56.8%, 57.8%, 28.9%, and 43.2% across our four time periods (Appendix A). One-dose RMR was similar in older and younger individuals. One-dose RMR, unlike two-dose RMR, did not exhibit waning. Thus, over time, the extra benefit from a second dose in reducing RMR, relative to one dose, has been shrinking.

### 3.8. Multivariate Estimates

In Table 3, we use a multivariate logistic model to predict RMRs for two-dose and three-dose vaccinees versus the unvaccinated. The multivariate model includes factors which are known predictors of COVID-19 mortality, including gender, race/ethnicity, education, and zip-SES [21,22]. Including these, additional predictors had little effect on RMR estimates. The multivariate RMRs are consistent with the rates presented in Table 2. For example, in 1H-2022 (Omicron period), multivariate RMR for all two-dose recipients versus the unvaccinated was 34.2%, versus 36.2% from Table 2.

### 3.9. Robustness Checks

Results for CEMP and RMR are consistent across a series of robustness checks: if we include the immune-compromised (Appendix A), exclude them and define immune-compromised more broadly (Appendix A), exclude J&J vaccine recipients (Appendix A), or use a shorter, 14-day minimum to treat vaccination as effective against death (Appendix A). The results were similar across genders and for Whites versus non-Whites (Appendix A).

In Appendix A, we estimate RMRs using the COVID-19 mortality rate as the outcome. RMRs measured this way are lower, consistent with the importance of the selection effects reported in Table 1. For example, for ages 60+, two-dose RMR across the sample periods was (8.3%, 10.1%, 16.6%, 24.6%) with COVID-19 mortality as the outcome, versus our finding above (11.1%, 20.8%, 27.7%, 34.2%).

## 4. Discussion

A central need, when estimating how vaccination affects COVID-19 mortality, is to estimate the counterfactual: what would COVID-19 mortality have been for the vaccinated, if they had not been vaccinated? We used non-COVID-NMR as a proxy for background mortality risk. We found important differences in background mortality risk between vaccinated and unvaccinated persons and between two-dose and three-dose recipients. Two-dose vaccinees are generally healthier (have lower non-COVID-NMR) than the unvaccinated, and three-dose vaccinees are generally healthier than two-dose vaccinees. These selection effects, unless controlled for (through our CEMP measure or in another way), can produce large biases in VE estimates.

Non-COVID-NMR performed well predicting COVID-19 mortality during the pre-vaccine period. We show this out-of-sample in Indiana in Figure 1, and in-sample, in Appendix A, for Wisconsin as a whole and Milwaukee County. This suggests that using CEMP as the outcome when measuring RMR provides a good estimate of the protective effects of vaccination relative to the counterfactual of mortality for the same persons, if they had been unvaccinated or received fewer doses. The similarity between the simple RMR estimates in Table 2, which control only for age group and time period, and the multivariate estimates in Table 3, provides further evidence that non-COVID-NMR, which we use in the CEMP denominator, did a good job of controlling for underlying health and mortality risk.

The data on non-COVID-NMR ratios in Table 1 can be used to assess the extent of selection bias: Assume counterfactually that vaccination was useless against COVID-19 mortality. What RMRs would one estimate, controlling only for age group and time period? Given the high correlation between non-COVID-NMR and COVID-19 mortality for the unvaccinated, the RMR ratios of vaccinated to unvaccinated persons in Table 1 provide approximate answers to this question. For example, for three-dose recipients aged 60–79 during the Omicron period, the estimated three-dose RMR would be 35.4%, even though there was, by assumption, no true vaccine effect.

### 4.1. Advantages of the CEMP Measure

CEMP, as a measure of COVID-19 mortality, has attractive features relative to other measures. It relies only on death certificates, which are available for all decedents, yet can address selection effects by using non-COVID-NMR to proxy for population health, which is otherwise difficult to observe. An alternative approach, controlling for comorbidities captured in electronic health records, faces important limitations: comorbidity data may not be fully reported, and, in the U.S., population-level data on comorbidities is not available. Even studies that control for comorbidities often examine only people who seek medical care for COVID-19 infection [8,21]. This will miss the association among underlying health, who becomes infected, and infection severity.

Using data only on decedents also avoids the challenges in estimating the population at risk. Population statistics may undercount some groups because of non-participation in government surveys (in the U.S., the Census or the American Community Survey), or inaccurate data. Race/ethnicity can be inaccurately captured in death certificate data, but this will not bias comparisons between different racial/ethnic groups unless the inaccuracies differ systematically between those who die of COVID-19 versus other natural causes.

### 4.2. Overview of Results: Substantial RMRs, Large Value for Boosters

Our analysis provides a number of insights with regard to the extent to which COVID-19 vaccines reduce mortality and for selection effects in who gets vaccinated. First, our two-dose RMRs versus unvaccinated persons are substantially higher (VE is lower) than in most other studies. The higher RMRs reflect our use of CEMP to address selection bias, as well as continued vaccine waning in the Omicron period.

The studies covered by the available systematic reviews report lower two-dose RMRs, from 6 to 17%, as compared to this study [1,2]. One study of U.S. veterans finds an 18% RMR for fully vaccinated U.S. veterans (two mRNA or one J&J) for ages <65 and 28% for ages 65+, in the pre-booster period [14]. This study used the rich VA data to control for comorbidities; it also found that vaccinees have lower all-cause mortality. A study of Sweden through January 2022 which controlled for health reported waning beginning 5 months after the second dose, with RMR (averaged across all ages) rising to 56.4% for 38–45 weeks after the second dose [23]. Other studies had few or no controls for selection effects. A study of Czechia through November 2021 reported two-dose RMR versus unvaccinated persons, 7–8 months after vaccination, as 17% for Pfizer and 12% for Moderna (2022), averaged across all ages [3]. A study of Greece through November 2021 reported that RMR rose to around 20% at 6 months after second dose, but studies only three broad age groups [4]. A study of Hungary reported two-dose RMR versus the unvaccinated that rose steadily with age [9]. A study of Norway over July–November 2021 reported two-dose RMR against death, averaged over both mRNA vaccines, as being 6.6% after 10–17 weeks, increasing to 31.4% for 33+ weeks [16]. The first three studies reported high near-term protection from a booster dose. A study of Puerto Rico through mid-October 2021 reported two-dose RMR after 144 days (longest period considered) of 14% for Pfizer and 7% for Moderna. Studies that compare three-dose to two-dose protection report substantial protection from a booster dose, although without the decomposition by age we provide in this study [11,12].

Second, we found substantial waning of two-dose protection against mortality, with two-dose RMR versus the unvaccinated for ages 60+ increasing from 11.1% in 2Q-2021 to 34.2% in 1H-2022. Note that we could not separate the effects of waning over time from differences in protection against different virus variants.

Third, we found that two-dose RMR increased with age, but boosters provide substantial additional protection. This makes them especially important for older persons. For ages 60+, three-dose RMR versus the unvaccinated was 9.5% in 4Q-2021 and 10.8% in 1H-2022, versus 27.7% and 34.2% for two-doses. At the same time, we found higher three-dose RMRs than reported in prior booster studies [10]. These higher values are consistent with the importance of controlling for selection effects. Three-versus-two-dose differences in RMRs for ages 60+ are large, at 18.2% in 4Q-2021 and 23.4% in 1H-2022. The reduction in RMR is even higher for ages 80+, at 32.2% for 1H-2022. In effect, the higher two-dose RMRs that we found, compared to most prior studies, leave more room for boosters to reduce mortality, even though we also find higher three-dose RMRs than prior research. Our evidence supports public health messaging and policy that encourages boosters for the elderly.

Fourth, we found stronger relative two-dose protection for ages 18–59 in 2021, compared to older persons in the pre-Omicron period, but not in the Omicron period. Absolute COVID mortality risk after two doses is smaller for younger persons, but boosters are effective in reducing that risk. We found zero deaths among three-dose recipients aged 18-59. Our results for the Omicron period contrast to the perception among many younger persons that two doses provide sufficient protection.

Fifth, we found that a single dose provides only moderate protection, with RMR versus the unvaccinated being around 50% (Appendix A). However, this protection appears to be long-lived. Limited waning has been reported before for the single-dose J&J vaccine [7,24]. We found similar results for one-dose mRNA recipients.

We lacked a sufficient sample size to separate the sample into narrower age groups. However, our point estimates suggest substantial booster value for ages 50–59, who suffered all six of the deaths among two-dose vaccinees, within the 40–59 age group, compared to no deaths for ages 50–59 among booster recipients.

### 4.3. An Opportunity for Targeted Booster Messaging

Evidence of vaccine waning first appeared in mid-2021, initially from Israel. Based on this evidence, Israel launched a booster campaign in late July 2021, which reached the whole population by the end of August [25]. Other countries soon followed, relying in part on Israeli evidence that boosters added important value. In the U.S., however, FDA scientists publicly questioned the need for boosters [26]. An advisory committee to the Food and Drug Administration (FDA) in September 2021 approved only a limited rollout to the elderly and persons at risk due to occupational exposure [27]; similarly, an advisory committee to the U.S. Centers for Disease Control (CDC) endorsed boosters only for the elderly [28].

Two months later, the FDA and CDC approved boosters for all adults; although a CDC recommendation came only at the end of November, 2021 [29]. However, U.S. public health messaging on boosters remained muddled, with the value of boosters “lost in the sea of changing recommendations and guidance.” [30]. Even today, U.S. booster percentages lag behind many other countries [9,31], and public knowledge of booster recommendations is limited [32]. Our study provides strong evidence on booster value for ages 60+, which account for the vast majority of COVID-19 deaths, and nearly all vaccinee deaths.

### 4.4. Toward Enhanced Public Reporting of COVID-19 Mortality

Many public sources report data on COVID-19 deaths. However, none reports a comparison to other natural deaths. Reporting both COVID-19 and non-COVID natural deaths, and ideally CEMP (the ratio of the two), would provide valuable information to the public on the risk of death from COVID-19 versus other natural causes. Reporting CEMP would show that the unvaccinated face a substantial increase in mortality risk due to COVID-19, even at younger ages. This might make the large reductions in mortality risk from vaccination more salient for younger persons. Reporting both COVID-19 mortality and other natural mortality by age group and number of vaccine doses received could also focus attention on selection effects, and their importance when estimating vaccine effectiveness.

## 5. Limitations

This study has important limitations. We studied only mortality, not other important adverse outcomes including hospitalization and long-term symptoms. COVID-19 mortality is uncommon for younger persons, which limits statistical power. Our data is only for Milwaukee County, which is racially, ethnically, and economically diverse, but its COVID-19 experience may not be representative of other areas.

We did not observe, and thus cannot control for, prior COVID-19 infection. Especially in the Omicron era, many people, both vaccinated and unvaccinated, were already infected. For them, VE can be understood as measuring the extra protection from hybrid resistance (from prior infection plus vaccination) versus natural resistance alone (from prior infection).

We did not observe individual health characteristics, except through the limited lens of death certificates. There could be differences between vaccinated and unvaccinated persons that affect COVID-19 mortality, which are not reflected in non-COVID-NMR.

The CEMP measure has several inherent limitations. Though likely a reasonable proxy for overall health, it does not consider behavioral or other differences between more and less vaccinated groups. Behavioral differences are likely to exist. For example. the unvaccinated or less vaccinated may believe COVID is less severe than the maximally vaccinated, and therefore take fewer precautions. Conversely, vaccinated persons may accept greater risks of becoming infected, because they believe they are protected against serious illness. Neither ours nor other VE studies can control for behavioral differences.

The CEMP measure assumes that COVID-19 infection does not meaningfully affect non-COVID mortality. Yet, COVID-19 infection is known to predict higher post-infection mortality from other causes, at least in the near term [33,34]. This will cause downward bias in CEMP values. If this bias is similar for the vaccinated and unvaccinated, RMR estimates should still be unbiased. The downward bias in CEMP could be larger for the unvaccinated, who will on average face more severe COVID-19. If so, then our RMR estimates will be somewhat below those we would estimate if we could attribute these extra natural deaths to COVID-19.

COVID infection could accelerate death for sick persons, especially the frail elderly, who may be near the end of their life in any case. This culling effect implies lower non-COVID-NMR for the unvaccinated following the peak COVID mortality period of late 2021–early 2022, but will not bias our RMR estimates.

The CEMP measure also assumes that vaccination does not meaningfully affect non-COVID mortality. Separating selection effects from any long-term effects of vaccination on non-COVID mortality is empirically challenging. However, studies of mRNA vaccine safety have found minimal evidence of near-term vaccine-related mortality. The main risks appear to be myocarditis and pericarditis for young men, generally mild and rarely fatal [35,36], and allergic reactions to the vaccination, rarely fatal if treated [37,38].

COVID-19 deaths could be underreported, but we coded COVID-19 deaths based on reading death certificates; this produced significantly larger counts than we would have obtained if we had relied on ICD-10 codes from the NCHS. Any remaining undercount appears small (Figure 2).

## 6. Conclusions

We used a novel outcome measure, CEMP, to study how vaccination affects COVID-19 mortality risk. This measure uses mortality from other natural causes to control for selection effects in who gets vaccinated. We found substantially lower non-COVID natural mortality risk for vaccinated than for unvaccinated persons. Thus, the vaccinated would likely face lower COVID-19 risk even if not vaccinated. After controlling for these selection effects, we found substantial vaccine protection against death, but also increasing two-dose RMR over time, and large differences in RMR after two doses between younger (age 18–59) and older (age 60+) people. These findings imply that boosters are highly important in reducing mortality, especially for ages 60+. The RMRs after two-dose vaccination, and the meaningful although smaller three-dose RMRs for ages 60+, imply that non-vaccine mitigation strategies remain an important tool in reducing mortality in vaccinated populations, particularly among the elderly.

## Figures and Tables

**Figure 1 vaccines-11-00379-f001:**
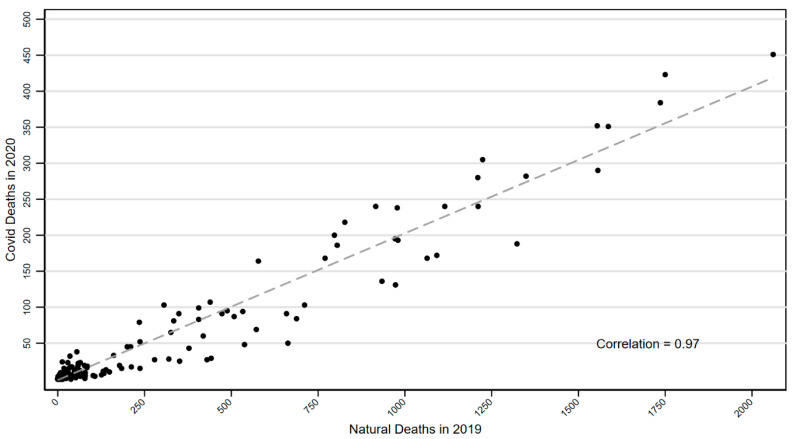
Out-of-Sample Correlation for Indiana between Natural Mortality in 2019 and COVID-19 Mortality in 2020. Figure shows scatterplot of natural mortality in Indiana over April–December 2019 against COVID-19 mortality over April–December 2020, for groups defined by age (18–39, 40–49, 50–59, 60–69, 70–79, 80–89, 90+), gender, race/ethnicity, and zip-SES quintile. This figure also shows a best-fit regression line and a Pearson correlation coefficient. See Appendix A for similar in-sample scatterplots for Wisconsin and Milwaukee County.

**Figure 2 vaccines-11-00379-f002:**
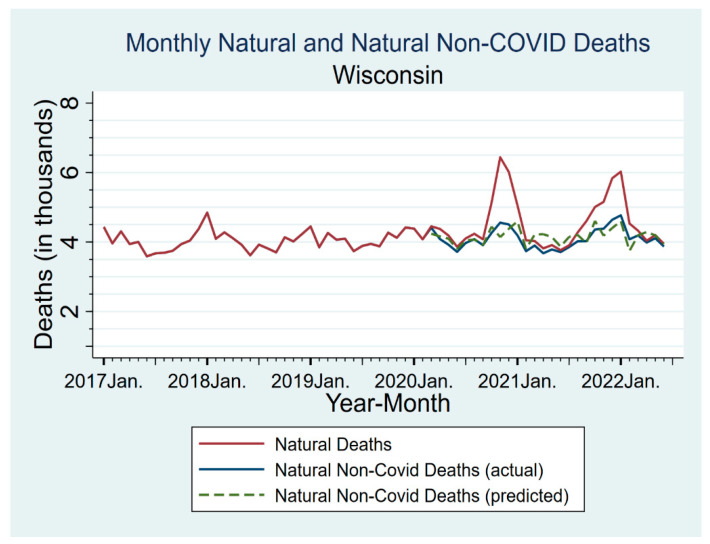
Actual versus Predicted Non-COVID Natural Mortality in Wisconsin. Figure shows monthly data for natural non-COVID and all natural deaths, for Wisconsin for January 2017–June 2022. For the pandemic period starting March 2020, we show both actual and predicted natural non-COVID deaths. Predicted deaths are based on linear extrapolation from 2017–2019 to the same calendar month during the pandemic period. Natural deaths (including COVID-19 deaths) are shown as solid red line; this shows two large COVID-related peaks in late 2020 and late 2021–early 2022. Natural non-COVID deaths (all natural deaths minus COVID-19 deaths) are shown as solid blue line. Predicted natural non-COVID deaths are shown as dashed green line.

**Figure 3 vaccines-11-00379-f003:**
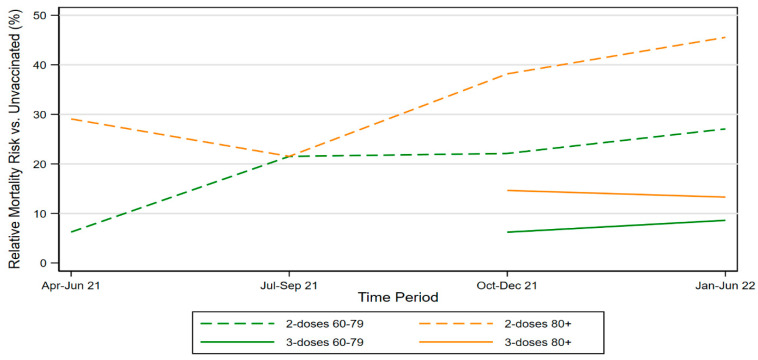
RMR for Two-Dose and Three-Dose Elderly Vaccinees. Figure shows two-dose and three-dose relative mortality risk (RMR) versus the unvaccinated, by time period, for persons aged 60–79 and aged 80+.

**Table 1 vaccines-11-00379-t001:** Non-COVID Natural Mortality Rate (non-COVID-NMR) by Age Group and Time Period.

	Time Period	April–June 2021 (Alpha)		Jul–Sep 2021 (Delta No Booster)		Oct–Dec 2021 (Delta, with Booster)			Jan–June 2022 (Omicron)		
Age	Measure	Unvax	2 doses	Unvax	2 doses	Unvax	2 doses	3 doses	Unvax	2 doses	3 doses
18–39	Population	197,089	86,870	149,102	133,762	124,348	145,124	12,796	104,434	106,970	73,509
	Non-COVID NMR	0.016%	0.001%	0.022%	0.007%	0.021%	0.008%	0.000%	0.031%	0.015%	0.008%
	ratio to unvax		**5.4% ****		**34.4% ****		**36.9% ****	**0.0% ^na^**		**49.0% ***	**27.1% ****
40–59	Population	113,674	88,215	82,177	122,364	65,647	119,938	30,562	56,743	73,873	81,111
	Non-COVID NMR	0.133%	0.053%	0.138%	0.046%	0.155%	0.077%	0.056%	0.313%	0.149%	0.110%
	ratio to unvax		**39.4% *****		**33.2% *****		**49.9% *****	**36.4% *****		**47.5% *****	**35.0% *****
60–79	Population	44,765	95,518	31,873	114,459	27,001	86,624	48,677	24,587	35,677	92,478
	Non-COVID NMR	0.852%	0.279%	0.930%	0.298%	0.984%	0.438%	0.198%	1.695%	1.194%	0.599%
	ratio to unvax		**32.7% *****		**32.0% *****		**44.5% *****	**20.1% *****		**70.4% *****	**35.4% *****
80+	Population	10,621	21,411	9502	23,475	8864	17,984	11,499	8004	6817	18,406
	Non-COVID NMR	2.582%	1.598%	1.988%	1.875%	2.550%	2.467%	1.032%	3.848%	4.743%	4.085%
	ratio to unvax		**61.9% *****		94.3%		96.7%	**40.5% *****		**123.3% ***	106.2%

Sample is adult decedents in Milwaukee County, Wisconsin, excluding immune-compromised persons. Table shows Non-COVID Natural Mortality Rate (NCNMR) and relative NCNMR versus the unvaccinated for people vaccinated with 2 doses or 3 doses. NCNMR is defined as the number of non-COVID-19 natural deaths occurring among a group of persons within an indicated age group with the indicated vaccination status over the indicated period, divided by the estimated population of people in the same group. Persons in each group were determined as of the first of each month, lagging vaccination dates by 14 days, and averaged over the indicated periods. For NCNMR ratios, *, **, *** indicates *p* < 0.05, 0.01, and 0.001, respectively, versus null of 100% (no difference relative to unvaccinated). Significant results (at *p* < 0.05 or better) are in boldface. Statistical significance cannot be assessed for the one cell with 0% NCNMR (indicated with na). Population counts are averaged over the months in each period.

**Table 2 vaccines-11-00379-t002:** CEMP and Relative Mortality Risk (RMR) by Age Group and Time Period.

		April–June 2021 (Alpha)		Jul–Sep 2021 (Delta No Booster)		Oct–Dec 2021 (Delta, with Booster)			Jan–June 2022 (Omicron)		
Age	Measure	**0 Doses**	**2 Doses**	**0 Doses**	**2 Doses**	**0 Doses**	**2 Doses**	**3 Doses**	**0 Doses**	**2 Doses**	**3 Doses**
18–39	COVID deaths	2	0	7	0	16	0	0	4	1	0
	Other natural deaths	31	1	32	9	26	11	0	32	18	4
	CEMP	6.5%	0.0%	21.9%	0.0%	61.5%	0.0%	NA	12.5%	5.6%	0.0%
	RMR vs. Unvax		0.0%		0.0%		0.0%	NA		44.4%	0.0%
40–59	COVID deaths	7	0	31	1	57	2	0	23	6	0
	Other natural deaths	147	27	113	53	102	95	3	177	109	84
	CEMP	4.8%	0.0%	27.4%	1.9%	55.9%	2.1%	0.0%	13.0%	5.5%	0.0%
	RMR vs. Unvax		0.0%		6.9%		3.8%	0.0%		42.4%	0.0%
60–79	COVID deaths	26	1	49	12	95	30	1	90	25	10
	Other natural deaths	367	226	297	338	266	380	45	416	427	537
	CEMP	7.1%	0.4%	16.5%	3.6%	35.7%	7.9%	2.2%	21.6%	5.9%	1.9%
	RMR vs. Unvax		6.2%		21.5%		22.1%	6.2%		27.1%	8.6%
80+	COVID deaths	6	2	26	13	49	37	2	56	27	18
	Other natural deaths	273	313	189	439	226	447	63	307	325	742
	CEMP	2.2%	0.6%	13.8%	3.0%	21.7%	8.3%	3.2%	18.2%	8.3%	2.4%
	RMR vs. Unvax		29.1%		21.5%		38.2%	14.6%		45.5%	13.3%
**Total 18–59**	COVID deaths	9	0	38	1	73	2	0	27	7	0
	CEMP	5.1%	0.0%	26.2%	1.6%	57.0%	1.9%	0.0%	12.9%	5.5%	0.0%
	RMR vs. Unvax		0.0% ^na^		**6.2% ****		**3.3% *****	0.0% ^na^		42.7%	0.0% ^na^
**Total 60+**	COVID deaths	32	3	75	25	144	67	3	146	52	28
	CEMP	5.0%	0.6%	15.4%	3.2%	29.3%	8.1%	2.8%	20.2%	6.9%	2.2%
	RMR vs. Unvax		**11.1% *****		**20.8% *****		**27.7% *****	**9.5% *****		**34.2% *****	**10.8% *****
**All**	**COVID deaths**	41	3	113	26	217	69	3	173	59	28
	**CEMP**	5.0%	0.5%	17.9%	3.1%	35.0%	7.4%	2.7%	18.6%	6.7%	2.0%
	**RMR (versus unvax)**		**10.6% *****		**17.3% *****		**21.1% *****	**7.7% *****		**36.2% *****	**11.0% *****

Table shows COVID deaths, natural non-COVID deaths, COVID Excess Mortality Percentage (CEMP), and relative mortality risk (RMR) for persons vaccinated with 2 or 3 doses, versus the unvaccinated. RMR for a comparison of two groups is the ratio of CEMPs for the two groups. Sample is adult decedents in Milwaukee County, Wisconsin, excluding immune-compromised persons. Due to the nature of the sample, CEMP ratios and RMRs are effectively weighted by mortality rates. Statistical significance for RMR ratios to unvaccinated is indicated for broad age groups (18–59; 60+); **, *** indicates *p* < 0.01, and 0.001, respectively versus null of 100% (no vaccine effect). Significant results (at *p* < 0.05 or better) are in boldface. Statistical significance is not reported for cells with 0% RMR (indicated with na).

**Table 3 vaccines-11-00379-t003:** COVID-19 Relative Mortality Risk (RMR) Calculated Using a Multivariate Logit Model.

			2 Doses			3 Doses		
Sample	Period	Remaining Risk	Estimate	*p*-Value	95% CI	Estimate	*p*-Value	95% CI
**18–59**	Apr–Jun 2021	Vs. unvax	0% ^na^			No booster		
	Jul–Sep 2021	Vs. unvax	**3.3% ****	**0.005**	**[0.3%, 36.3%]**	No booster		
	Oct–Dec 2021	Vs. unvax	**2.4% *****	**0.000**	**[0.6%, 9.9%]**	0% ^na^		
		Vs. 2 doses				0% ^na^		
	Jan–Jun 2022	Vs. unvax	**36.7% ***	**0.041**	**[14.0%, 96.1%]**	0% ^na^		
		Vs. 2 doses				0% ^na^		
**60+**	Apr–Jun 2021	Vs. unvax	11.7% ***	0.001	[3.4%, 40.3%]	No booster		
	Jul–Sep 2021	Vs. unvax	21.7% ***	0.000	[13.2%, 35.6%]	No booster		
	Oct–Dec 2021	Vs. unvax	**27.9% *****	**0.000**	**[20.1%, 38.7%]**	**9.6% *****	**0.000**	**[3.0%, 30.7%]**
		Vs. 2 doses				33.9%	0.068	[10.6%, 108.2%]
	Jan–Jun 2022	Vs. unvax	**32.9% *****	**0.000**	**[23.5%, 46.0%]**	**11.1% *****	**0.000**	**[7.2%, 17.1%]**
		Vs. 2 doses				**31.5% *****	**0.000**	**[19.1%, 51.9%]**
**All (18+)**	Apr–Jun 2021	Vs. unvax	**11.9% *****	**0.001**	**[3.5%, 40.4%]**	No booster		
	Jul–Sep 2021	Vs. unvax	**19.0% *****	**0.000**	**[12.0%, 30.2%]**	No booster		
	Oct–Dec 2021	Vs. unvax	**21.9% *****	**0.000**	**[16.2%, 29.5%]**	**8.2% *****	**0.000**	**[2.6%, 25.9%]**
		Vs. 2 doses				33.5%	0.065	[10.5%, 107.1%]
	Jan–Jun 2022	Vs. unvax	**34.2% *****	**0.000**	**[25.0%, 46.8%]**	**10.3% *****	**0.000**	**[6.7%, 15.8%]**
		Vs. 2 doses				**29.2% *****	**0.000**	**[18.1%, 47.2%]**

Table shows the odds ratios from logit regressions, for COVID-19 death as the outcome, for sample of persons who died of natural causes. Odds ratios are for two- and three dose vaccinees, relative to unvaccinated persons, and for three-dose vaccinees relative to two-dose vaccinees, during the indicated time periods. These odds ratios directly measure RMRs. Odds ratios are from logit model of Prob (COVID-19 Death) = f (doses received, baseline is unvaccinated or two-dose vaccinated depending on the RMR being estimated), with controls for age, age-squared, zip-SES (measured in centiles), gender, race/ethnicity, education level, marital status, and military veteran status. 95% confidence intervals (CIs) are in parentheses. Sample is same as Table 2. Coefficients on covariates are suppressed. *, **, *** indicates *p* < 0.05, 0.01, and 0.001, respectively, versus null of 100% (no vaccine effect). Significant results (at *p* < 0.05 or better) are in boldface. Statistical significance is not reported for cells with 0% RMR (indicated with na).

## Data Availability

The linked mortality and vaccination data on which this study relies was obtained under a data use agreement with the Wisconsin Department of Health Services and cannot be publicly shared.

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
