# Peer review of "Understanding COVID-19 Vaccine Effectiveness against Death Using a Novel Measure: COVID Excess Mortality Percentage"

_vaccines, 2023, doi:10.3390/vaccines11020379_

Round 1
Reviewer 1 Report
Dear authors
This is an interesting study about an important and current subject. The study is very interesting and is a further contribution to understanding COVID-19 Vaccine Effectiveness.
Comments:
It is essential to clarify ethical issues in the methods; the authors should mention the ethics committee that approved the study, as well as whether the study complies with data protection regulations.
. Introduction:
The purpose of the study should be further clarified. It is important in the introduction to review the literature by including more references at European level.
. Methods:
The methods have several limitations, many of which are described in the discussion, but some issues should be clarified, for example: - in the choice of age range, the 18-39 range can be very different; - another issue that should be properly explained is the difference in counting results between the research team and NCHS. To clarify the ???? it is suggested that the researchers of this work, present a flowchart with the group (????? ????ℎ?) and with group (n??−????? ??????? ????ℎ?).
. Results:
In table 1, it is important to identify the N, for each of the age groups.
In table 3, it is not clear why age ranges different from those used in the previous tables were chosen.
Discussion:
It would be important to discuss the results obtained through the questionnaires with other studies conducted in other countries on this issue.
Author Response
Reviewer 1 Substantive Comments
This is an interesting study about an important and current subject. The study is very interesting and is a further contribution to understanding COVID-19 Vaccine Effectiveness.
Comments:
It is essential to clarify ethical issues in the methods; the authors should mention the ethics committee that approved the study, as well as whether the study complies with data protection regulations.
Reply. A statement on institutional review board (IRB) approval was included at the end of the prior draft. We moved this statement to the end of Part 2 on Data and Methods. In our experience as U.S.-based authors, it is not customary to state separately that the research plan complies with privacy regulations, because this is encompassed by IRB review and approval. We can do so, of course, if requested.
Introduction:
The purpose of the study should be further clarified. It is important in the introduction to review the literature by including more references at European level.
Reply. There are several aspects to this question. One is whether we had reviewed and cited the most relevant European studies. We had done so in the prior draft, but had not discussed individual studies. We now discuss these studies. A second is where to place the more detailed literature review that you requested – in the introduction or the discussion section. We thought it made more sense to discusses the principal studies, including the Europe-based studies, in the discussion section.
We added details on five population-level studies of European countries, three of which we had already cited but not specifically discussed in the references: Lytras et L. (2022) (Greece); Berec et al. (2022) (Czechia), and Kiss et al. (2022) (Hungary). We chose not to cite or rely on a followup study of boosters during the Omicron period by the Hungarian group because they counted as a COVID-19 death all deaths of COVID-positive persons, regardless of actual cause of death. This is problematic in the Omicron period because many hospitalizations are of persons “with COVID,” who were hospitalized primarily for other reasons. The approach in this study to defining COVID death is not ideal during 2021 either, but we judged that it was acceptable for the pre-Omicron period, compared to omitting discussion of this study.
We also updated our literature review, and found two additional studies: Xu et al. (2022, Sweden), and Starrfelt et al. (2022, Norway). These are now also cited and discussed. We also added details in the Appendix on the criteria we used to decide which studies we viewed as worth citing.
Methods:
The methods have several limitations, many of which are described in the discussion, but some issues should be clarified, for example: - in the choice of age range, the 18-39 range can be very different.
Reply. There were too few deaths in the 18-39 age group to allow us to carry out further decomposition. There was only one COVID death among vaccinated people in this age group. Thus, we cannot assess whether vaccine effectiveness differs between, say ages 18-29 and ages 30-39.
For similar reasons, we did not feel comfortable separating the age 40-59 group into ages 40-49 and ages 50-59. We added a comment to the text on the reasons why we did not use narrower age groups, and a statement that for ages 50-59, substantial risk remains after two-doses, and boosters likely provide important protection.
another issue that should be properly explained is the difference in counting results between the research team and NCHS.
Reply. The Appendix provides details on our approach, and documents the differences. We added some additional discussion of the reasons for the differences to the text, but were limited in what we could say by space constraints.
To clarify the ???? it is suggested that the researchers of this work, present a flowchart with the group (????? ????ℎ?) and with group (n??−????? ??????? ????ℎ?).
Reply. Thank you for this suggestion. We added a flowchart to the Appendix (new Figure App-1), and refer to it in the text.
Results:
In table 1, it is important to identify the N, for each of the age groups.
Reply. Done. This is a bit tricky because people get vaccinated over time. We provided averages for each of the time periods in our study, and explain this in the table heading.
In table 3, it is not clear why age ranges different from those used in the previous tables were chosen.
Reply. We reported only aggregated age ranges in Table 3 to reduce the complexity of this table. In our view, a principal added value of Table 3 was to show consistency of the multivariate estimates in Table 3 with the simpler estimates in Table 2. We have added to the Appendix, Table App-6, multivariate regression results for all age groups.
Discussion:
It would be important to discuss the results obtained through the questionnaires with other studies conducted in other countries on this issue.
Reply. As discussed above, we added details to the discussion section on other studies of vaccine effectiveness against mortality.

Reviewer 2 Report
The authors investigated the Covid-19 vaccine effectiveness using data on all deaths in the Milwaukee County, Wisconsin, from April 2020 to June 2022. The most important consideration when measuring vaccine effectiveness is the bias caused by the tendency for a larger proportion of originally healthy people to receive the vaccine. The authors introduced the COVID Excess Mortality Percentage (CEMP) to avoid this bias and compared CEMP in vaccinated and unvaccinated subjects. CEMP is the number of people who died in COVID-19 divided by the number of people who died otherwise. The main focus of this study is on the relative mortality risk (RMR), which is the CEMP of the vaccinated divided by the CEMP of the unvaccinated. If the RMR is smaller than 1, the vaccine is considered effective. Table 2 summarizes the estimated value of RMR for the vaccination status (no, 2-dose, 3-dose) at the four time periods and for four age groups. The authors claim the effectiveness of the vaccine based on small RMR values, saying in particular “Our evidence supports public health messaging and policy that encourages boosters for the elderly.”
However, the authors overlook an important possibility. That is, one of the reasons for the smaller RMR could be that vaccinations increase natural mortality (i.e., increase CEMP in the denominator of RMR). In fact, as the two columns in the lower right of Table 1 indicate, the elderly over 80 years of age are more likely to die “naturally” after vaccination. If, as the authors say, many people who are vaccinated are originally healthy, this death may be caused by adverse reactions to vaccines. If this possibility cannot be ruled out, it is a gesture of the mass murderer to say “Our evidence supports public health messaging and policy that encourages boosters for the elderly.”
I would recommend that this manuscript be published only after improvements have been made based on the above and the following comments.
Minor
1) Page 1, line 23 (Abstract): r=0.97 does not make sense.
2) Page 5, line 185: The values 0.038% and 0.008% are supposed to be the sum of the NCNMR ratios for the first two periods, but I think the ratios should not be added together.
3) Page 6, line 202: “as Table 1” is not correct in the caption of Table 1
4) Page 8, line 245: “age 18-49” should be “age 18-39.”
5) It would be better to unify the number of significant digits in the main text and that in tables
6) Page 8, lines 265-266: I don't know which part of the table is corresponding to the statement “Thus, over time, as the benefit from a second dose, in reducing RMR has been shrinking.”
7) Page 10, lines 320-321: I don't know which part of the table is corresponding to the statement “For example, for ages 40-79 during the Omicron period, estimated three-dose 320 RMR would be 35%, versus true RMR (no vaccine effect) was 100%.”
Author Response
Reviewer 2 Substantive Comments
The authors investigated the Covid-19 vaccine effectiveness using data on all deaths in the Milwaukee County, Wisconsin, from April 2020 to June 2022. The most important consideration when measuring vaccine effectiveness is the bias caused by the tendency for a larger proportion of originally healthy people to receive the vaccine. The authors introduced the COVID Excess Mortality Percentage (CEMP) to avoid this bias and compared CEMP in vaccinated and unvaccinated subjects. CEMP is the number of people who died in COVID-19 divided by the number of people who died otherwise. The main focus of this study is on the relative mortality risk (RMR), which is the CEMP of the vaccinated divided by the CEMP of the unvaccinated. If the RMR is smaller than 1, the vaccine is considered effective. Table 2 summarizes the estimated value of RMR for the vaccination status (no, 2-dose, 3-dose) at the four time periods and for four age groups. The authors claim the effectiveness of the vaccine based on small RMR values, saying in particular “Our evidence supports public health messaging and policy that encourages boosters for the elderly.”
However, the authors overlook an important possibility. That is, one of the reasons for the smaller RMR could be that vaccinations increase natural mortality (i.e., increase CEMP in the denominator of RMR). In fact, as the two columns in the lower right of Table 1 indicate, the elderly over 80 years of age are more likely to die “naturally” after vaccination. If, as the authors say, many people who are vaccinated are originally healthy, this death may be caused by adverse reactions to vaccines. If this possibility cannot be ruled out, it is a gesture of the mass murderer to say “Our evidence supports public health messaging and policy that encourages boosters for the elderly.”
Reply. Thank you for this comment. Given the evidence from other sources on vaccine safety, it had not occurred to us to address this possibility in the draft. But in fact, the results we present, and additional results added to the Appendix in response to your comment, go a long way toward ruling out the “vaccination increases natural non-COVID death” story. And a good thing too!
Our story involves selection bias, in which the vaccinated are generally healthier than the unvaccinated, which we measure and the control for using non-COVID natural death rates. But if vaccines are causing excess natural non-COVID deaths, then our measure of selection bias is incorrect.
The empirical challenge is that we observe non-COVID natural mortality for the vaccinated, but do not observe what their natural mortality would have been if they had not been vaccinated. In causal inference terms, we do not observe the missing potential outcome: non-COVID natural mortality for the vaccinated if they had not been vaccinated. The empirical challenge is compounded because if COVID deaths were undercounted, then the missed COVID deaths would show up as non-Covid natural deaths.
There is no direct escape from this problem. But we believe that we can rule out a large effect as follows. For the elderly, the great majority are vaccinated. Thus, if vaccination caused excess deaths among the vaccinated, we ought to be able to see this by comparing the observed non-Covid natural mortality rate for all persons (vaccinated or not) to the rate that would be expected by extrapolating the natural mortality rate from the pre-Covid period to the Covid period, and seeing whether there are excess non-Covid natural deaths. We can carry out this extrapolation only for all persons, because in the pre-Covid period, we cannot separate the mortality rate for persons who will be vaccinated in the future from that for persons who will remain unvaccinated.
As it happens, we carried out exactly this comparison in Figure 2, and provided confidence intervals in Appendix Figure App-3. We did so to address the objection that if COVID deaths were undercounted, the CEMP numerator would be too low, the denominator would be too high, and CEMP would be biased downwards. We did not have your concern in mind. However, Figure 2 should address your concern as well – and we say so in the revised draft. If vaccination caused excess natural non-COVID deaths, this should show up in measured non-COVID deaths exceeding predicted deaths. Moreover, since any effect of vaccination would likely fade over time, one would expect to find excess natural non-COVID deaths at the times when vaccination rates were high, in the first half of 2021.
This is not observed. Instead, predicted natural non-COVID natural deaths (dashed line in Figure 2) are close to measured non-COVID natural deaths (solid blue line), sometimes higher or lower, but with no obvious pattern. Measured non-COVID natural deaths are never above the 95% confidence interval (CI) for the predicted deaths (Appendix Figure App-3). In the first half of 2021, we observe the opposite: measured non-Covid natural deaths are below predicted, and sometimes below the bottom of the 95% CI for predicted natural deaths (Figure App-3).
To address the possibility that there might be an effect of vaccination for the often-frail elderly, especially ages 80+, that might not be apparent from a single graph covering all ages, we prepared similar graphs for ages 18-59, 60-79, and 80+, and include these in the revised Appendix. As you can see from Appendix Figure App-4, there is no excess natural non-COVID mortality either for ages 60-79 or for ages 80+, either overall or during the time period (first half of 2021) when any adverse effect of vaccination would be most likely.
Indeed, for ages 80+, who would be the most vulnerable to any shock from vaccination, we see the opposite in 1H-2021. Measured natural non-COVID death rates is below predicted, and indeed below the bottom of the 95% CI for predicted rates. This suggests the following story, which we view as worth exploring further, although it is outside scope for our current project: To some extent, COVID-19 accelerated death for the frail elderly, many of whom would have died fairly soon anyway. This inference is strengthened because we see the same pattern in 1H-2022, following the Delta-Omicron spike in Covid deaths in late 2021-early 2022.
In sum, we believe we have ruled out vaccines as a weapon for mass murder for the frail elderly. In responding to your comment, we have also found evidence suggesting a new area of inquiry: To what extent did COVID-19 deaths for the frail elderly accelerate deaths that would have happened soon anyway? Thank you! (Whether we want to publish a “Covid pushed the frail elderly over the edge” paper, given that it is not exactly politically correct, is a separate question.)
I would recommend that this manuscript be published only after improvements have been made based on the above and the following comments.
Minor
1) Page 1, line 23 (Abstract): r=0.97 does not make sense.
Reply. This is the Pearson correlation coefficient for Indiana between natural deaths in April-December, 2019 and COVID-19 deaths for the same months in 2020, which is also shown graphically in Figure 1. We spelled this out in the revision, instead of just saying “r=0.97.”. The Appendix includes similar figures for Wisconsin as a whole, and for Milwaukee County. The correlation coefficients are 0.98 for Wisconsin as a whole, and 0.94.
2) Page 5, line 185: The values 0.038% and 0.008% are supposed to be the sum of the NCNMR ratios for the first two periods, but I think the ratios should not be added together.
Reply. We removed the references to specific percentages in this paragraph. An alternative, which we can implement easily if requested, would be to provide “for example” percentages for one of the time periods in our study.
3) Page 6, line 202: “as Table 1” is not correct in the caption of Table 1
Reply: Fixed. Our apologies. We had flipped the order of Table 1 and Table 2 prior to submission, so that in the submitted version, Table 2 had the sample details, but overlooked the need to change the Table headings. Both tables now specify the sample.
4) Page 8, line 245: “age 18-49” should be “age 18-39.”
Reply. This statement was in fact true for ages 18-49, because there were no deaths among two-dose vaccinees aged 40-49. To avoid confusion, we changed the text to refer to ages 18-39, for which this statement is also true. We added a paragraph to the text discussing ages 50-59, who do have meaningful mortality after two vaccine doses.
5) It would be better to unify the number of significant digits in the main text and that in tables
Reply. Done, for numbers that also appear in the text tables. We left more rounded numbers (e.g., 74% rather than 74.2%) for other numbers, because we generally prefer not to include digits that do not have substantive importance.
6) Page 8, lines 265-266: I don't know which part of the table is corresponding to the statement “Thus, over time, as the benefit from a second dose, in reducing RMR has been shrinking.”
Reply. Rephrased. This referred to the additional protection from two doses, versus one dose. Note that, to reduce clutter in the tables in the text, we moved the one-dose results to appendix.
7) Page 10, lines 320-321: I don't know which part of the table is corresponding to the statement “For example, for ages 40-79 during the Omicron period, estimated three-dose RMR would be 35%, versus, true RMR (no vaccine effect) was 100%.”
Reply. For simplicity, we revised the text to refer to ages 60-79, for which the corresponding values are in Table 1.

Round 2
Reviewer 1 Report
Thank you for the corrections/amendments made.
It would be important to revise the English.
Author Response
Comments and Suggestions for Authors
Thank you for the corrections/amendments made.
It would be important to revise the English.
Reply. It was difficult for us to respond to this general comment because the reviewer provided no specific examples. The draft was reread and edited for clarity (at the cost of a bit of extra length) by the native-English speaking senior author (Black) and reviewed for clarity by a second native-English speaking coauthor (Meurer).
Reviewer 2 Report
Although my concerns about the negative aspects of vaccines have not been completely dispelled by reading the reply, I too hope that it will be published soon, as it contains valuable data for analyzing the effects of vaccines.
I understand, of course, that there is no period in Figure 1 where excess mortality is significantly greater, but my main concern is not about 2021, but about the trends after 2022.
As I mentioned in the first review comment, the bottom columns in Table 1 for the 80+ age group is a source of concern. The NCNMR ratio to un-vax in the first half of 2021 (61.9%) must mean that the vaccinated people were very healthy. I think the figures in the first half of 2021 are good for all age groups. However, by late 2021, the NCNMR ratio to un-vax is approximately 100%, which shows no bias that the vaccinators are healthy; one can see that after October, healthy people have started to receive their third vaccine. In 2022, the NCNMR ratio to un-vax has risen to 123.3%, which means that vaccinated people are starting to die at a higher rate. I am also concerned that the NCNMR ratio to un-vax for other age groups is also on the increase. In other words, is vaccination likely to lead to long-term ill-health rather than acute manifestations? This may be a naive view, but I think this part of Table 1 would discourage at least people over 80 years of age to take the vaccine unless they have presbyopia and cannot read the figures.
